# Quantifying gliding forces of filamentous cyanobacteria by self-buckling

Maximilian Kurjahn[1]*, Antaran Deka[1], Antoine Girot[1,2], Leila Abbaspour[3,4], Stefan Klumpp[3,4], Maike Lorenz[5], Oliver Bäumchen[1,2], Stefan Karpitschka[1,6]*

[1]Max Planck Institute for Dynamics and Self-Organization (MPI-DS), Göttingen, Germany; [2]Experimental Physics V, University of Bayreuth, Bayreuth, Germany; [3]Max Planck School Matter to Life, University of Göttingen, Göttingen, Germany; [4]Institute for Dynamics of Complex Systems, University of Göttingen, Göttingen, Germany; [5]Department of Experimental Phycology and SAG Culture Collection of Algae Albrecht-von-Haller Institute for Plant Science, University of Göttingen, Göttingen, Germany; [6]Fachbereich Physik, University of Konstanz, Konstanz, Germany

**Abstract** Filamentous cyanobacteria are one of the oldest and today still most abundant life-forms on earth, with manifold implications in ecology and economics. Their flexible filaments, often several hundred cells long, exhibit gliding motility in contact with solid surfaces. The underlying force generating mechanism is not yet understood. Here, we demonstrate that propulsion forces and friction coefficients are strongly coupled in the gliding motility of filamentous cyanobacteria. We directly measure their bending moduli using micropipette force sensors, and quantify propulsion and friction forces by analyzing their self-buckling behavior, complemented with analytical theory and simulations. The results indicate that slime extrusion unlikely generates the gliding forces, but support adhesion-based hypotheses, similar to the better-studied single-celled myxobacteria. The critical self-buckling lengths align well with the peaks of natural length distributions, indicating the importance of self-buckling for the organization of their collective in natural and artificial settings.

## eLife assessment

This **valuable** paper describes innovative force measurements of the bending modulus of gliding cyanobacteria, along with measurements of the critical buckling length of the cells, which in combination lead to insight into how these cells produce the force necessary to move. Quantitative analysis **convincingly** shows that the propulsive force and resistive friction coefficient are strongly coupled, which supports propulsion based on adhesion forces rather than slime extrusion.

## Introduction

Filamentous cyanobacteria are an omnipresent group of phototrophic prokaryotes, contributing majorly to the global fixation of atmospheric carbon dioxide. They played an important role already in the paleoclimate of our planet, having generated the atmospheric oxygen (*Whitton, 2012*; *Sciuto and Moro, 2015*) on which animal life is based. Today, giant marine and limnic blooms pose ecological and economical threats (*Whitton, 2012*; *Sciuto and Moro, 2015*; *Bauer et al., 2020*; *Fastner et al., 2018*), but also enable bioreactor applications, for instance as a renewable energy source (*Whitton, 2012*; *Ruffing and Kallas, 2016*). The long and flexible filaments contain up to several hundred, linearly stacked cells. Many species exhibit gliding motility when in contact with solid surfaces or other filaments, but no swimming motion (*Burkholder, 1934*). Motility enables filaments to aggregate into colonies, adapting their architecture to environmental conditions (*Whitton, 2012*; *Khayatan*

*For correspondence: maximilian.kurjahn@ds.mpg.de (MK); stefan.karpitschka@uni-konstanz.de (SK)

Sent for Review 22 March 2023
Preprint posted 27 April 2023
Reviewed preprint posted 06 October 2023
Reviewed preprint revised 01 May 2024
Version of Record published 12 June 2024

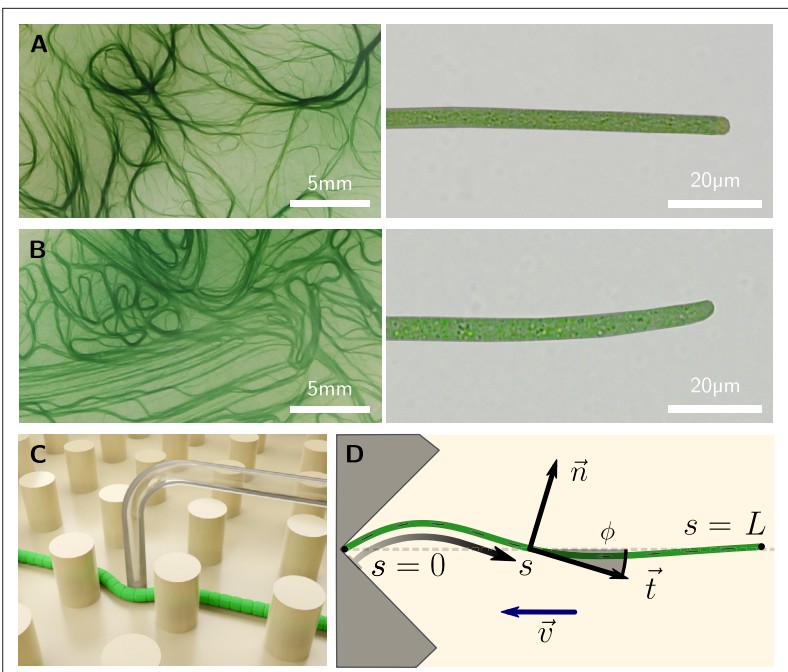

**Figure 1.** Colony and filament morphology of gliding filamentous cyanobacteria, bending, and buckling tests. (**A, B**) Colonies on agar plates (left) and individual filaments in liquid medium (right) of *K. animale* and *O. lutea*, respectively. (**C**) Schematic of a microscopic three-point bending test, pushing a filament into the gap between SU-8 pillars using a glass micropipette. (**D**) Schematic of a self-buckling test in a microfluidic chip: A filament glides into a V-shaped obstacle and buckles if its contour length $L$ exceeds the self-buckling threshold $L_c$.

*et al., 2015*). The force generating mechanism behind gliding is not yet understood (*Halfen and Castenholz, 1971*; *Godwin et al., 1989*; *Hoiczyk and Baumeister, 1998*; *McBride, 2001*; *Gupta and Agrawal, 2006*; *Read et al., 2007*; *Koiller et al., 2010*; *Hanada et al., 2011*; *Khayatan et al., 2015*; *Wilde and Mullineaux, 2015*). Slime extrusion (*Hoiczyk and Baumeister, 1998*), metachronal waves on surface fibrils (*Halfen and Castenholz, 1971*; *Halfen, 1973*; *Read et al., 2007*), and acoustic streaming (*Koiller et al., 2010*) have been proposed. A few species appear to employ a type-IV-pilus-related mechanism (*Khayatan et al., 2015*; *Wilde and Mullineaux, 2015*), similar to the better-studied myxobacteria (*Godwin et al., 1989*; *Mignot et al., 2007*; *Nan et al., 2014*; *Copenhagen et al., 2021*; *Godwin et al., 1989*), which are short, rod-shaped single cells that exhibit two types of motility: S (social) motility based on pilus extension and retraction, and A (adventurous) motility based on focal adhesion (*Chen and Nan, 2022*), for which also slime extrusion at the trailing cell pole was earlier postulated as mechanism (*Wolgemuth, 2005*). Yet, most gliding filamentous cyanobacteria do not exhibit visible pili and their gliding mechanism appears to be distinct from myxobacteria (*Khayatan et al., 2015*).

Here, we measure the bending moduli of *Kamptonema animale* and *Oscillatoria lutea* (*Figure 1A and B*, respectively) by micropipette force sensors (*Backholm and Bäumchen, 2019*). This allows us to quantify the propulsion and friction forces associated with gliding motility, by analyzing their self-buckling behavior. Self-Buckling is an important instability for self-propelling rod-like micro-organisms to change the orientation of their motion, enabling aggregation or the escape from traps (*Fily et al., 2020*; *Man and Kanso, 2019*; *Isele-Holder et al., 2015*; *Isele-Holder et al., 2016*). The notion of self-buckling goes back to work of Leonhard Euler in 1780, who described elastic columns subject to gravity (*Elishakoff, 2000*). Here, the principle is adapted to the self-propelling, flexible filaments (*Fily et al., 2020*; *Man and Kanso, 2019*; *Sekimoto et al., 1995*) that glide onto an obstacle. Filaments buckle if they exceed a certain critical length $L_c \sim (B/f)^{1/3}$, where $B$ is the bending modulus and $f$ the propulsion force density. By recording numerous collision events, we obtain a comprehensive statistics to derive $L_c$. Kirchhoff beam theory provides an analytical expression for the prefactor in $L_c$, and we numerically calculate the evolution of the filament shape upon buckling. Comparing experiment with theory, we derive the propulsion force densities and friction coefficients of the living

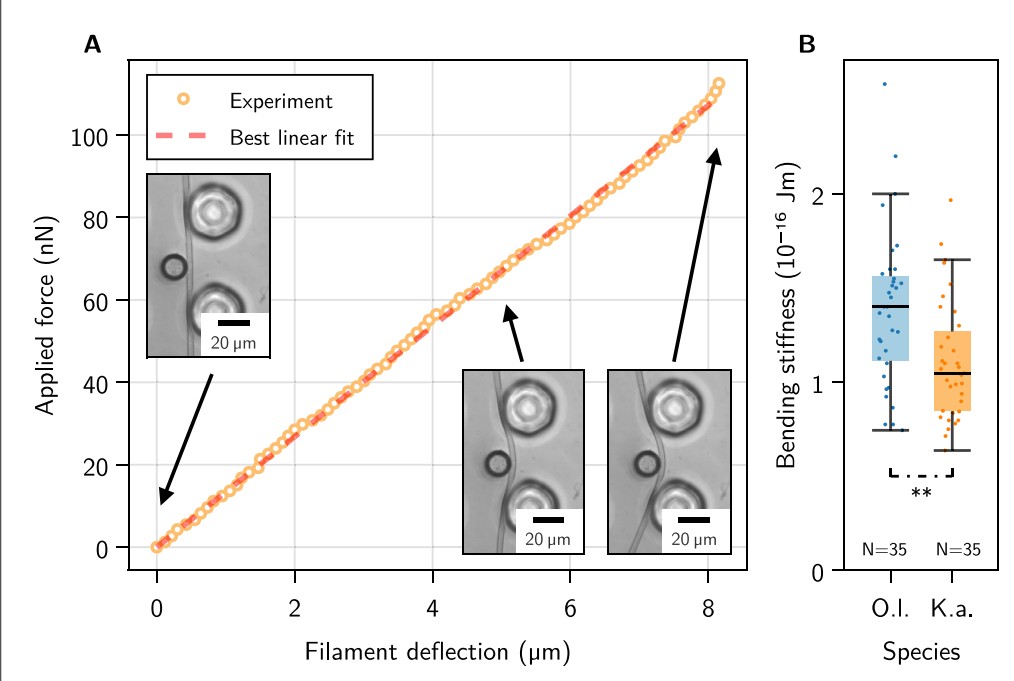

**Figure 2.** Single-filament bending measurements using micropipette force sensors. (**A**) Representative force-deflection measurement for *K. animale*. Insets: Bottom-view micrographs of the same experiment. (**B**) Box plot of the bending moduli for $N = 35$ individuals of *K. animale* and *O. lutea*, shown as points (each tested 2-10 times). Box limits denote the first and third quartile, whiskers the last measurement within the inter-quartile distance away from the respective box limit. A p-value of 6.87 suggests different typical moduli for the two species.

The online version of this article includes the following figure supplement(s) for figure 2:

**Figure supplement 1.** Details on the microscopic three-point bending tests.

filaments. Force and friction are strongly coupled, which favors an adhesion-based propulsion mechanism (***Khayatan et al., 2015***; ***Wilde and Mullineaux, 2015***) over the still customary slime-extrusion hypothesis (***Hoiczyk and Baumeister, 1998***; ***McBride, 2001***). The critical lengths we found are close to the peak in the length distribution in freely growing colonies, indicating the importance of this quantity for their self-organization.

## Results

### Bending measurements

The bending moduli $B$ of individual filaments of *O. lutea* and *K. animale* were measured by microscopic three-point bending tests (***Backholm et al., 2013***). Filaments that glide freely across liquid-immersed surfaces decorated with micro-pillars (SU-8 on glass) were pushed into a gap between two pillars with a Micropipette Force Sensor (MFS, see *Figure 1C*). The deflection of the lever arm of the micropipette is proportional to the load acting on its tip. The corresponding spring constant is obtained from independent calibration measurements (see Methods and *Figure 2—figure supplement 1A*). The base of the pipette was actuated with a constant speed of ±5 µm/s to increase and release the force acting on the living filament. Pipette and filament deflections were analyzed with a custom-made image analysis procedure in Matlab (see Methods and *Kreis et al., 2018*; *Kreis et al, 2019*; *Backholm and Bäumchen, 2019*; *Böddeker et al., 2020* for details).

 *Figure 2A* shows an exemplary force-displacement curve, accompanied with snapshots from the experiment, for *K. animale*. The measured force-distance relations were continuous, linear, largely speed-independent and free of hysteretic effects (*Figure 2—figure supplement 1B and C*), allowing for an analysis with standard beam theory to derive the effective bending modulus $B$ from the slopes of the force-deflection curves.

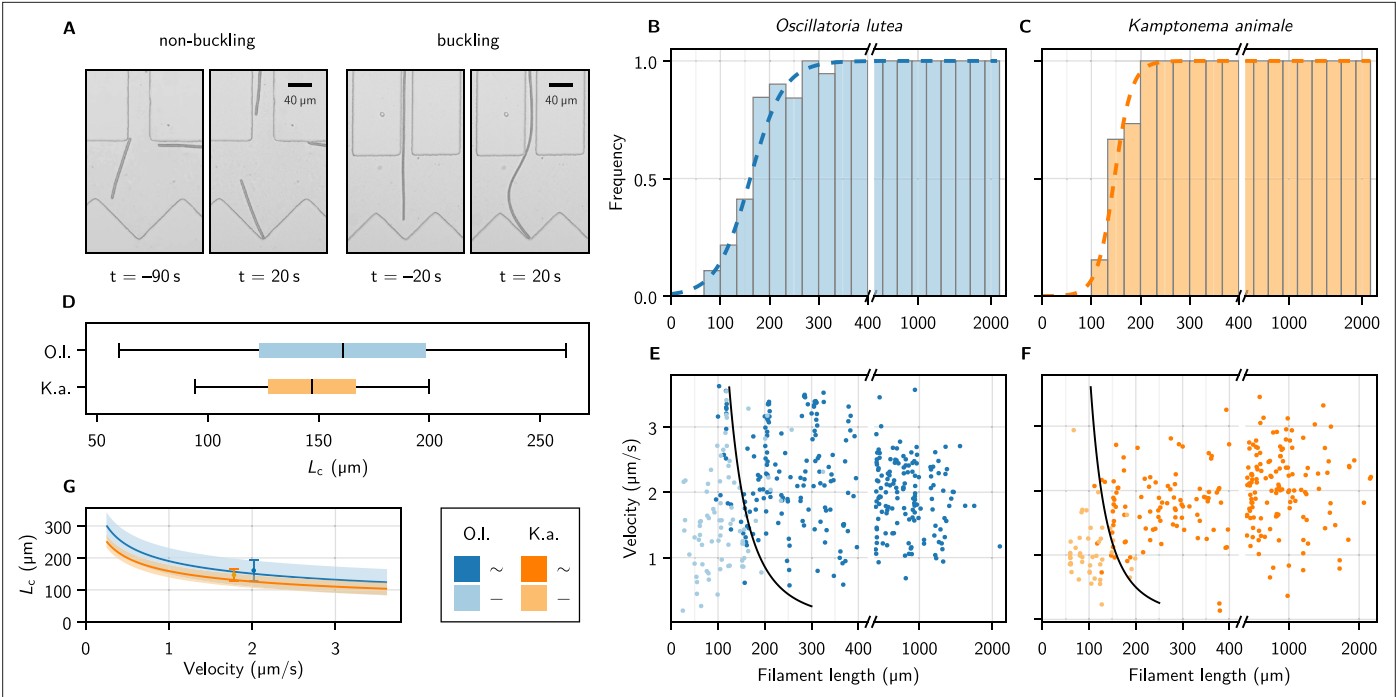

**Figure 3.** Self-buckling experiments and statistics. (**A**) Snapshots of *K. animale*, before and after hitting the obstacle at $t = 0$. Left, short filament with $L < L_c$, right, long filament with $L > L_c$. (**B, C**) Bar plot of the buckling frequency vs. filament length for *O. lutea* (**B**, totally $N = 388$ events) and *K. animale* (**C**, totally $N = 280$ events), together with the logistic regression (dashed curve). (**D**) Box plot of the quantiles of the critical length distribution from the logistic regression $p(L)$. Box limits denote first and third quartiles, whiskers the 5th and 95th percentile. (**E, F**) Velocity $v_0$ immediately before hitting the obstacle vs. filament length $L$ for *O. lutea* and *K. animale*, respectively, distinguishing buckling (dark) and non-buckling (light). The velocity-dependent median critical length $L_c(v_0)$, as derived from a logistic regression with $L$ and $v_0$ as independent explanatory variables, is indicated by black lines. Note that axes in (**B, C, E, F**) are broken around $L = 2L_c$ to emphasize the critical region. (**G**) $L_c(v_0)$ (lines) and inter-quartile region (shaded), together with the simple logistic regression from (**D**), located at the mean velocity $\bar{v}_0$ of the population (symbols & error bars).

The online version of this article includes the following figure supplement(s) for figure 3:

**Figure supplement 1.** Collection of buckling events with different shapes of the traps.

**Figure supplement 2.** Natural length distributions and buckling probability.

**Figure supplement 3.** Simulations of self-buckling semi-flexible chains of active overdamped particles.

Each individual filament was tested two to ten times at different locations along its contour (see *Figure 2—figure supplement 1D* for a collection of individual measurements). We observed no systematic dependence of $B$ on the length of the filament or the position along the contour, i.e., no softening or stiffening toward the ends. Stiffness variations on scales below the pillar spacing can of course not be excluded. *Figure 2B* shows a box plot of the bending moduli. *O. lutea* appears to be slightly stiffer with $B = (1.4 \pm 0.4) \times 10^{-16}$ Jm than *K. animale* with $B = (1.0 \pm 0.3) \times 10^{-16}$ Jm.

## Buckling measurements

We now turn from micropipette bending measurements to self-buckling experiments. The buckling behavior was observed by optical microscopy in quasi-two-dimensional microfluidic compartments filled with liquid medium (*Figure 1D*, *Figure 3A*, and Methods). The height of the chambers was approximately 5 µm, only slightly larger than the diameter of the filaments, such that motion and buckling was confined to the $x$-$y$-plane. Filaments explored the entire device and occasionally entered channels that directed them onto V-shaped traps (opening angle 90°). The V-shape is not necessarily required but reduces the chance of filaments slipping sideways instead of buckling, as was observed sometimes for collisions with flat walls. For a collection of collision events with various obstacle architectures, see *Figure 3—figure supplement 1*. After colliding, the filaments escaped these traps, either by reversing their gliding direction or, if they buckled, due to the reorientation of their front. In total, we collected 388 collision events for *O. lutea* and 280 for *K. animale*.

The observed events were classified as *buckling* or *non-buckling* manually by visual inspection (*Figure 3A* and *Figure 3—figure supplement 1*). We observed no systematic dependence of the buckling behavior on the shape and size of the trap, nor on the angle of incidence. The filament length $L$ as well as the free gliding velocity $v_0$ prior to hitting the obstacle were determined by automated image processing (see Methods). Buckling frequencies (*Figure 3B and C*, bars) were evaluated by binning the observations into fixed intervals of the contour length $L$. Frequently, individual filaments were observed $N$ times, and previous buckling behavior is not readily repeated. Multiple observations of an individual filament were weighted with $1/N$ to obtain an unbiased representation of the population.

The weighted events were analyzed by a logistic regression of the buckling probability

$$p = \text{sig}(x) = \left(1 + e^{-x}\right)^{-1}, \tag{1}$$

with $x = (L - L_c)/\Delta L_c$. The median critical length $L_c$ and the width of its distribution $\Delta L_c$ are obtained by maximum likelihood estimation (Methods). The results are depicted as the dashed curves in *Figure 3B and C*. For *O. lutea*, we find $L_c \pm \Delta L_c = (161 \pm 35)$ µm and for *K. animale* $(148 \pm 18)$ µm. The corresponding box plot is shown in *Figure 3D*.

The substrate contact requires lubrication from polysaccharide slime to enable bacteria to glide (*Khayatan et al., 2015*). Thus we assume an over-damped motion with co-linear friction, for which the propulsion force $f$ and the free gliding velocity $v_0$ of a filament are related by $f = \eta v_0$, with a friction coefficient $\eta$. In this scenario, $f$ can be inferred both from the observed $L_c \sim (f/B)^{-1/3}$ and, up to the proportionality coefficient $\eta$, from the observed free gliding velocity. Thus, by combining the two relations, one may expect also a strong correlation between $L_c$ and $v_0$. In order to test this relation for consistency with our data, we include $v_0$ as a second regressor, by setting $x = (L - L_c(v_0))/\Delta L_c$ in *Equation 1*, with $L_c(v_0) = (\eta v_0/(30.5722 B))^{-1/3}$, to reflect our expectation from theory (see below). Now, $\eta$ rather than $f$ is the only unknown, and its ensemble distribution will be determined in the regression. *Figure 3E and F* show the buckling behavior as color code in terms of the filament length $L$ and the free gliding velocity $v_0$ prior to hitting the obstacle. From maximum likelihood estimation of $L_c(v_0)$ (black lines), we obtain $\eta = (0.6 \pm 0.4)\,\text{nN}\,\text{s}\,\mu\text{m}^{-2}$ for *O. lutea* and $\eta = (0.8 \pm 0.6)\,\text{nN}\,\text{s}\,\mu\text{m}^{-2}$ for *K. animale*. In *Figure 3G*, we compare $L_c(v_0)$ for both species, and the results from the one-parameter regression, placed at the mean velocity $\bar{v}$.

Within the characteristic range of observed velocities (1–3 µm s$^{-1}$), the median $L_c$ depends only mildly on $v_0$, as compared to its rather broad distribution, indicated by the bands in *Figure 3G*. Thus a possible correlation between $f$ and $v_0$ would only mildly alter $L_c$. The natural length distribution (*Figure 3—figure supplement 2*), however, is very broad, and we conclude that growth rather than velocity or force distributions most strongly impacts the buckling propensity of cyanobacterial colonies. Also, we hardly observed short and fast filaments of *K. animale*, which might be caused by physiological limitations (*Burkholder, 1934*).

## Buckling theory

In the classical self-buckling theory by Euler, the critical length for a vertical column, clamped at its lower end and free at its upper end, of uniform bending modulus $B$, subject to a gravitational force density $f_g$, is given by $L_c = \left(7.837 B/f_g\right)^{1/3}$(*Elishakoff, 2000*). The buckling of gliding filaments differs in two aspects: the propulsion forces are oriented tangentially instead of vertically, and the front end is supported instead of clamped. Therefore, with $L < L_c$ all initial orientations are indifferently stable, while for $L > L_c$, buckling induces curvature and a resultant torque on the head, leading to rotation (*Fily et al., 2020*; *Chelakkot et al., 2014*; *Sekimoto et al., 1995*). Buckling under concentrated tangential end-loads has also been investigated in literature (*De Canio et al., 2017*; *Wolgemuth, 2005*), but leads to substantially different shapes of buckled filaments.

Following existing literature (*Fily et al., 2020*; *Chelakkot et al., 2014*; *Sekimoto et al., 1995*), we use classical Kirchhoff theory for a uniform beam of length $L$ and bending modulus $B$, subject to a force density $\vec{b} = -f\vec{t} - \eta\vec{v}$, with an effective active force density $f$ along the tangent $\vec{t}$, and an effective friction proportional to the local velocity $\vec{v}$. Presumably, this friction is dominated by the lubrication drag from the contact with the substrate, filled by a thin layer of secreted polysaccharide slime which is much more viscous than the surrounding bulk fluid. Speculatively, the motility mechanism might also comprise adhering elements like pili (*Khayatan et al., 2015*) or foci (*Mignot et al., 2007*)

that increase the overall friction (**Pompe et al., 2011**). Thus, the drag due to the surrounding bulk fluid can be neglected (**Man and Kanso, 2019**), and friction is assumed to be isotropic, a common assumption in motility models (**Fei et al., 2020**; **Tchoufag et al., 2019**; **Wada et al., 2013**). We assume a homogeneous and constant distribution of an effectively tangential active force along the filament. Since many cells contribute simultaneously to the gliding force, one may expect noise and fluctuations on the length scale of the individual cell, far below the filament length, which we thus neglect. Based on our observations, we assume a planar configuration and a vanishing twist. Thus we also neglect any helical components of active force or friction, since these appear to merely add rigid-body rotation during free gliding. We parametrize the beam by its orientational angle $\phi(s)$ as a function of the contour coordinate $s$ (see **Figure 1D**), to obtain the Kirchhoff equation (**Audoly and Pomeau, 2018**)

$$B\,\partial_s\kappa - \vec{n}\cdot\int_s^L ds'\,\left(f\vec{t}(s') + \eta\,\vec{v}(s')\right) = 0, \tag{2}$$

with $\kappa = \partial_s\phi$, the curvature and $\vec{n}$, the unit normal vector. The head of the filament ($s = 0$) is subject to a localized force $\vec{P}$ that balances the load integral and thereby fixes its position. The tail ($s = L$) is naturally force-free, and the two boundary conditions are vanishing torques at the head and tail of the filament:

$$\kappa|_{s=0} = \kappa|_{s=L} = 0. \tag{3}$$

The local velocity is expressed through

$$\vec{v} = \partial_t\vec{x} = \partial_t\int_0^s ds'\,\vec{t}(s') = \int_0^s ds'\,\vec{n}(s')\,\partial_t\phi(s'). \tag{4}$$

Inserting **Equation 4** into **Equation 2** and changing the order of integration, the inner integral can be evaluated to obtain

$$B\,\partial_s^2\phi - \vec{n}\cdot\left\{\int_s^L ds'\,\left\{f\vec{t} + \eta\,(L-s')\,\vec{n}\,\partial_t\phi\right\} + \eta\,(L-s)\int_0^s ds'\,\vec{n}\,\partial_t\phi\right\} = 0. \tag{5}$$

**Equation 5** is solved by the method of lines (see Methods and **Figure 4—figure supplement 2**).

To derive the critical self-buckling length, **Equation 5** can be linearized for two scenarios that lead to the same $L_c$: early-time small amplitude buckling and late-time stationary rotation at small and constant curvature (**Fily et al., 2020**; **Chelakkot et al., 2014**; **Sekimoto et al., 1995**). Scaling $s$ by $L$ and $t$ by $t_0 = L^4\eta/B$, a single dimensionless parameter remains, the activity coefficient $\Gamma = L^3 f/B$, reminiscent of the flexure number typically used in statistical physics of active polymers (**Isele-Holder et al., 2015**; **Isele-Holder et al., 2016**). Seeking stationary rotor solutions $\phi(s,t) = \phi(s) + \omega\,t$, rotating with angular frequency $\omega$, **Equation 5** reduces to (in scaled units) (**Chelakkot et al., 2014**; **Fily et al., 2020**; **Sekimoto et al., 1995**)

$$\partial_s^2\kappa + \Gamma\,(1-s)\,\kappa + \omega\,s = 0. \tag{6}$$

This second order ordinary differential equation is subject to three boundary conditions, $\kappa|_{s=0} = \kappa|_{s=1} = \partial_s\kappa|_{s=1} = 0$, hence fixing $\omega(\Gamma)$. The trivial solution $\kappa \equiv 0$ with $\omega = 0$ is amended by a first non-trivial branch of buckled filaments at $\Gamma \approx 30.5722$, the root of a combination of hypergeometric functions which is given in Methods and was first found by **Sekimoto et al., 1995**. Thus, in physical units, the critical length is given by $L_c = \left(30.5722\,B/f\right)^{1/3}$, which is reproduced in particle-based simulations (**Figure 3—figure supplement 3**) analogous to those in **Isele-Holder et al., 2015**; **Isele-Holder et al., 2016**.

Inserting the population median and quartiles of the distributions of bending modulus and critical length, we can now quantify the distribution of the active force density for the filaments in the ensemble from the buckling measurements. We obtain nearly identical values for both species, $f \sim (1.0 \pm 0.6)\,\mathrm{nN}/\mu\mathrm{m}$, where the uncertainty represents a wide distribution of $f$ across the ensemble rather than a measurement error.

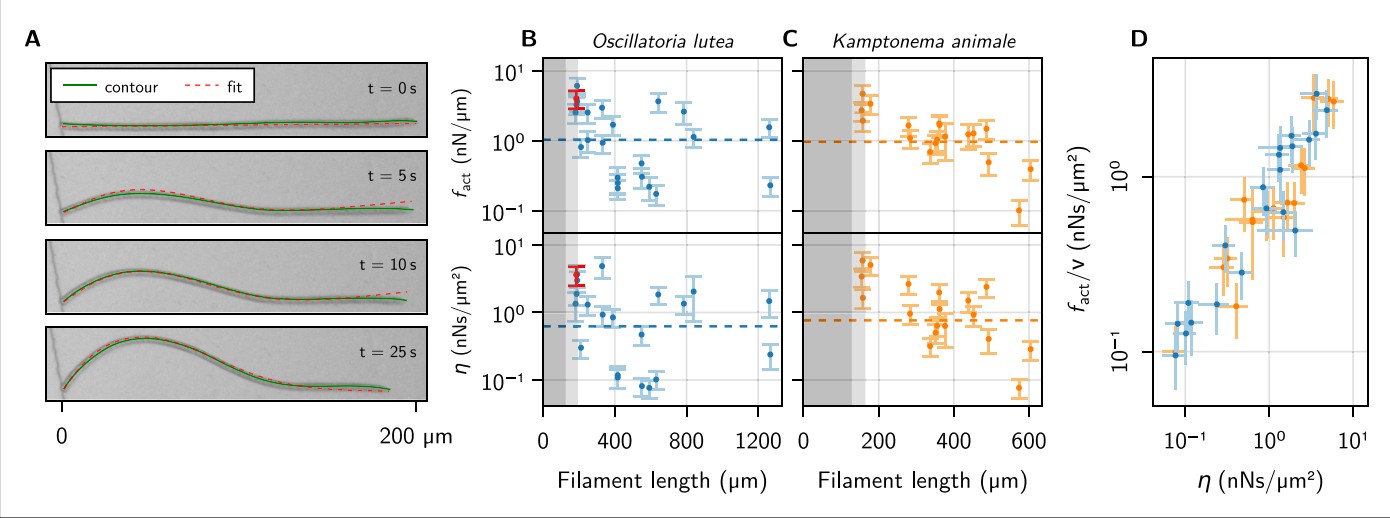

**Figure 4.** Kirchhoff theory compared to experimental contours. (**A**) Time series of a characteristic buckling event of *O. lutea*, overlaying the contour extracted from the images (green) and the best fit of the solution to *Equation 5* (red). (**B, C**) $f_{act}$ and $\eta$ vs. filament length $L$, as determined from the fit, for *O. lutea* and *K. animale*, respectively. Light gray indicates $L \in L_c \pm \Delta L_c$ (see *Figure 3D*), dark gray $L < L_c - \Delta L_c$, where buckling is not observed. The dashed lines are obtained from $L_c$ of logistic regression. The filament from (**A**) is indicated in red. (**D**) Free-gliding friction coefficient $f_{act}/v$ against $\eta$ from the buckling profile fit.

The online version of this article includes the following figure supplement(s) for figure 4:

**Figure supplement 1.** Active force $f_{act}$ and friction coefficient $\eta$ vs. filament velocity.

**Figure supplement 2.** Numerical results for the full non-linear Kirchhoff theory.

## Profile analysis

We will now compare the evolution of theoretical profiles from *Equation 5* to the evolution of the experimental buckling contours. The latter were extracted from the micrographs with an in-house trained convolutional neural network with modified U-Net architecture (*Ronneberger et al., 2015*) (see *Figure 4A*, green contour) and tracked from the moment of impact until parts other than the head made contact with the confining walls. This limitation narrows down the available data substantially because buckling frequently induced contact with the channel walls. *Figure 4A* shows a representative time series of a buckling filament, together with the extracted contour and the fitted solution of *Equation 5* (see Methods for details of the fitting procedure). From the fit, we calculate $f$ and $\eta$ of individual filaments, using the median bending modulus of the respective species (*Figure 4B and C*). The dashed line on each panel indicates the value from the logistic regression, representative of the population. Filaments below the critical length do not buckle and no values can be derived from profile fitting: this region is indicated in gray. The light gray zone corresponds to the central quartiles of the critical length distribution, where data are biased toward larger forces because only part of the population can buckle. Indeed, here we record the strongest filaments. This bias is not present in the logistic regression, which is most sensitive in the transition region but equally accounts for buckling and non-buckling outcomes.

## Discussion

Remarkably, the median active forces for the two species match almost perfectly. The logistic regression gives a good estimate for the population average, and the distributions derived from individual profile fits are centered around this median. The similarity between the two species indicates a potential homology of their gliding apparati.

The comparison with Kirchhoff theory allows us to measure active forces and friction coefficients on an individual basis, going beyond the population mean. Thus it allows for a more insightful analysis, correlating, for instance, these values with length and free gliding speeds. We see no significant correlation between $L$ or $v_0$ and $f$ or $\eta$, but the observed values of $f$ and $\eta$ cover a wide range (*Figure 4B and C* and *Figure 4—figure supplement 1*). This is consistent with the logistic regression,

where using $v_0$ as a second regressor did not significantly reduce the width of the distribution of critical lengths or active forces. The two estimates of the friction coefficient, from logistic regression and individual profile fits, are measured in (predominantly) orthogonal directions: tangentially for the logistic regression where the free gliding velocity was used, and transversely for the evolution of the buckling profiles. Thus, we plot $flv$ over $\eta$ in *Figure 4D*, finding nearly identical values over about two decades. Since $f$ and $\eta$ are not correlated with $v_0$, this is due to a correlation between $f$ and $\eta$. This relation is remarkable in two aspects: On the one hand, it indicates that friction is mainly isotropic. This suggests that friction is governed by an isotropic process like bond friction or lubrication from the slime layer in the contact with the substrate, the latter being consistent with the observation that mutations deficient in slime secretion do not glide but exogenous addition of slime restores motility (*Khayatan et al., 2015*). In contrast, hydrodynamic drag from the surrounding bulk fluid (*Man and Kanso, 2019*), or the internal friction of the gliding apparatus would be expected to generate strongly anisotropic friction. If the latter was dominant, a snapping-like transition into the buckling state would be expected, rather than the continuously growing amplitude that is observed in experiments. On the other hand, it indicates that friction and propulsion forces, despite being quite variable, correlate strongly. Thus, generating more force comes, inevitably, at the expense of added friction. For lubricated contacts, the friction coefficient is proportional to the thickness of the lubricating layer (*Snoeijer et al., 2013*), and we conjecture active force and drag both increase due to a more intimate contact with the substrate. This supports mechanisms like *focal adhesion* (*Mignot et al., 2007*) or a *modified type-IV pilus* (*Khayatan et al., 2015*), which generate forces through contact with extracellular surfaces, as the underlying mechanism of the gliding apparatus of filamentous cyanobacteria: more contacts generate more force, but also closer contact with the substrate, thereby increasing friction to the same extent. Force generation by slime extrusion (*Hoiczyk and Baumeister, 1998*), in contrast, would lead to the opposite behavior: More slime generates more propulsion, but also reduces friction. Besides fundamental fluid-mechanical considerations (*Snoeijer et al., 2013*), this is rationalized by two experimental observations: i. gliding velocity correlates positively with slime layer thickness (*Dhahri et al., 2013*) and ii. motility in slime-secretion deficient mutants is restored upon exogenous addition of polysaccharide slime. Still, we emphasize that many other possibilities exist. One could, for instance, postulate a regulation of the generated forces to the experienced friction, to maintain some preferred or saturated velocity.

Finally, we remark that the distribution of $L_c$ aligns well with the peak of natural length distributions (see *Figure 3—figure supplement 2*). Dwelling in soil or as floating aggregates, natural colonies experience less ideal geometries than in our experiments. Nonetheless, we expect a similar buckling behavior since gliding requires contact with a surface or other filaments. As a consequence, small changes in the propulsion force density or the length distribution determine whether the majority of the filaments in a colony is able to buckle or not. This, in turn, has dramatic consequences on the exploration behavior and the emerging patterns (*Isele-Holder et al., 2015*; *Isele-Holder et al., 2016*; *Abbaspour et al., 2023*; *Duman et al., 2018*; *Prathyusha et al., 2018*; *Jung et al., 2020*): $(L/L_c)^3$ is, up to a numerical prefactor, identical to the flexure number (*Isele-Holder et al., 2015*; *Isele-Holder et al., 2016*; *Duman et al., 2018*; *Winkler et al., 2017*), the ratio of the Péclet number and the persistence length of active polymer melts. Thus, the ample variety of non-equilibrium phases in such materials (*Isele-Holder et al., 2015*; *Isele-Holder et al., 2016*; *Prathyusha et al., 2018*; *Abbaspour et al., 2023*) may well have contributed to the evolutionary success of filamentous cyanobacteria.

## Methods
### Cell cultivation
Species *Oscillatoria lutea* (SAG 1459–3) and *Kamptonema animale* (SAG 1459–6) were obtained from The Culture Collection of Algae at Göttingen University and seeded in T175 culture flask with standard BG-11 nutrition solution. The culture medium was exchanged for fresh BG-11 every four weeks. Cultures were kept in an incubator with an automated 12 hr day (30% light intensity (~20 µE), 18°C) and 12 hr night (0% light intensity, 14°C) cycle, with a continuous 2 hr transition. All experiments were performed at a similar daytime to ensure comparable phases in their circadian rhythm. The night cycle began at 11 a.m. so experiments could typically be started in the morning towards the end of the bacteria's day.

## Bending measurements

Rectangular arrays of cylindrical micropillars (base diameter 35 μm and pitch of 80 μm in both directions) were fabricated using standard SU-8 photolithography on transparent glass wafers. A liquid sample chamber was made by placing a rectangular microscope glass slide on top of the pillar-decorated substrate, using an O-ring cut into two pieces as spacers. After filling the chamber with standard BG-11 nutrient solution, a small fragment from the cyanobacterial cultures is introduced into the chamber with a syringe.

As filaments dispersed into the pillar-decorated surface autonomously by their gliding motility, a small region with sparsely distributed individual filaments is chosen for bending measurements. Then, a filament is bent between two pillars (about 6 orders of magnitude stiffer than the filament) with the nozzle of the L-bent glass micropipette force sensor (spring constant 9.5 ± 0.3 nN/μm), mounted on a motorized linear actuator (Newport Corporation, LTA-HS). The speed at which the nozzle moves as well as the amplitude of its displacement is set by the actuator. A detailed procedure for the micropipette fabrication and calibration can be found in *Backholm and Bäumchen, 2019*. In brief, the micropipette is calibrated by measuring the corresponding cantilever deflection under the applied weight of an evaporating water droplet hanging at the pipette nozzle (see *Figure 2—figure supplement 1A*). The corresponding error on the pipette spring constant is given by the standard deviation after independent subsequent calibration measurements. We note that this calibration method provides the spring constant of the micropipette in the direction of the nozzle, while during the bending measurements, a sideways deflection is used. We assume that the elastic properties of the pipette cantilever are isotropic, and therefore consider the spring constant of sideways deflection to be equal to the calibration direction.

Image sequences of deflections were recorded at 20× magnification and 40 fps with an Olympus IX-83 inverted microscope and a scientific CMOS camera (PCO Edge 4.2). The images were then analyzed with a custom-made image analysis procedure in Matlab, to determine the deflections of the filament and the pipette simultaneously. The deflection of the micropipette is obtained by subtracting the time-dependent position of the piezo controller, which is actuating the base of the pipette, from the nozzle position in the image. The force exerted by the pipette is given by its spring constant times its deflection (Hooke's law, see *Backholm and Bäumchen, 2019*). We note that no torsion of the micropipette cantilever was observed while bending the filament, thus, we assume that torsional modes do not play a significant role in the micropipette deflection analysis. The obtained data of the applied force and filament deflection results in a linear plot as shown in *Figure 2A*.

We derived the bending modulus according to standard beam theory, through $B = \frac{\Delta x^3}{48} \partial P / \partial d$, where $\Delta x$ is the distance between the pillars, $P$ is the force provided through the pipette, at the center between the two pillars, and $d$ is the deflection of the filament. Our values are comparable but slightly larger than values recently derived from cross-flow drag experiments (*Faluweki and Goehring, 2022*).

*Figure 2—figure supplement 1* shows results for different speeds, increasing and decreasing force, as well as a set of exemplary force-distance curves.

## Buckling experiments

We prepared microfluidic devices according to standard procedures of SU-8 clean-room photolithography, followed by PDMS-based soft lithography (*Fujii, 2002*), binding the cured PDMS imprints to rectangular glass coverslip by plasma activation (Electronic Diener Pico plasma system, air, 50% exposure, 30 s). Prior to binding, two 1 mm holes for flushing the device with BG-11 medium, and one 2 mm hole for loading cyanobacteria to the device, were punched in the PDMS, keeping the punch-outs for sealing the chip later on. Four different device architectures, each with 20 μm and 40 μm wide channels, with heights of ~ 5 μm were used.

The devices were first flushed with approximately 5 μL of conditioned BG-11 medium, through one of the small ports. Then, about 1mm³ of blue-greenish cyanobacteria were loaded into the device through the large port. Finally, the device was sealed with the cylindrical stoppers retained from the punching, and covered by a small round cover slip to minimize evaporation from the device during the experiment.

Buckling experiments were observed by a Nikon Ti2-E inverted microscope on a passive anti-vibration table, with transmitted illumination at about 20 μE illumination intensity. Microscopy images

were taken at 6x- or 10x-magnification with time intervals of either 1 s, 10 s, and 30 s for a couple of hours at a resolution of 4096 × 4096 pixels with a CMOS camera (Dalsa Genie Nano XL).

## Image analysis of buckling events

Regions of interest were cropped from the image sequences for each of the manually detected collision events, from 50 s before to 50 s after the collision, and analyzed further. The length of each filament was obtained by manually adding a path on top of the images in a standard image editor (Gimp). The decision of whether a filament buckles or not is made manually by watching the video of each event. The velocity is determined by extracting the position of the head of a filament prior to hitting the obstacle for up to six snapshots, and taking the mean traveled distance over this period.

Profiles were extracted only for selected events in which no additional collisions of the filament with the confining walls were observed for at least 10 s after the first contact of the head with the obstacle. First, the microscopy images were processed by an in-house trained, modified U-Net to detect their mid-lines (contours). These contour representations of the images were then vectorized into subpixel-accurate $x$-$y$-coordinates, to obtain the green contour from *Figure 4A*.

## Logistic regression

We perform a logistic regression on the individual (weighted) buckling events with a maximum likelihood estimation. This classification algorithm approximates the probability distribution by a logistic function; see *Equation 1*. By maximizing the log-likelihood, we find the parameters that best predict the buckling probability. The likelihood of correctly predicting the buckling ($y = 1$) or non-buckling ($y = 0$) behavior of a filament of known length $L$ is given by

$$P(y|L) = \left[\text{sig}\left(\frac{L - L_c}{\Delta L_c}\right)\right]^y \cdot \left[1 - \text{sig}\left(\frac{L - L_c}{\Delta L_c}\right)\right]^{1-y}, \tag{7}$$

with two parameters $L_c$ and $\Delta L_c$ that describe the median critical length and the width of the distribution, respectively. Each individual $i$ with length $L_i$ is observed $N_i$ times to determine the buckling outcomes $y_{i,j}$. The log-likelihood for representing all the data by a logistic distribution is then given by

$$\log \mathcal{L}(L_c, \Delta L_c) = \sum_{i,j} \frac{y_{i,j}}{N_i} \log \text{sig}\left(\frac{L_i - L_c}{\Delta L_c}\right) + \frac{1 - y_{i,j}}{N_i} \log \left[1 - \text{sig}\left(\frac{L_i - L_c}{\Delta L_c}\right)\right], \tag{8}$$

where the weight of each observation is given by $1/N_i$, the number of observations of individual $i$, to yield an unbiased estimate for the subsample of the population. Maximal $\mathcal{L}$ requires vanishing derivatives of $\log \mathcal{L}$ with respect to the parameters $L_c$, $\Delta L_c$. For the regression with two explanatory variables $L$ and $v_0$ i.e., $L_c(v_0) = (\alpha \, v_0)^{-1/3}$, the same procedure is used, adding the derivative with respect to $\alpha$ to the minimization criteria.

## Numerics and fitting

The evolution of the contour shapes according to *Equation 5* was derived for 30 different values of $\Gamma$, ranging from just above the critical length up to $L/L_c \sim 9$, by a numerical solution of *Equation 5*. *Equation 5* was discretized into $n = 64$ segments, defining the discrete $\phi_i$ on the midpoints of the intervals. Second order polynomial interpolation was then used to evaluate differential and integral terms. Time integration was performed with the method of lines, initialized with a solution to the linearized small-amplitude equation. Snapshots of the solution were stored for 64 times, ranging from small amplitude to head angles $\phi(s = 0) \sim 90°$. These profiles were linearly interpolated in $s$ and $t$ to obtain a continuous function for fitting to the experimental contours.

As the residual for fitting theoretical profiles to the experiments, we used the square distance between the experimental and theoretical profiles, integrated along the contour. The activity coefficient $\Gamma$ and time scale $t_0$, together with a rotational and two translational degrees of freedom, were then adapted to minimize the sum of the residuals. First, the theoretical profile with the smallest mean square distance is determined individually for each frame in an experimental time series, with simulation time, $L_c$, rotation, and translation as free parameters. The average over these individual fit results were then used as initial parameters for a global fit, where the sum of the residuals of all time steps was minimized simultaneously to derive a global parameter set, containing the time scale $t_0$, a

time offset, the critical length $L_c$, rotation, and translation. In order to estimate the error of the fit, we applied a very coarse bootstrapping, repeating the fit 20 times with randomly chosen subsets of the time steps.

For molecular dynamics simulations of buckling, the filaments were discretized as chains of $N$ beads with diameter $\sigma$ and a distance $\sigma/2$ between consecutive beads. Flexibility is implemented with a harmonic bending potential, $U_b = \frac{\kappa_b}{2} \sum_{j=2}^{N-1} (\theta_j - \pi)^2$, where $\theta_j$ is the angle between consecutive beads $i-1, i, i+1$, and self-propulsion by an active force $F_a$ on each bead, oriented tangentially along the chain (**Abbaspour et al., 2023**). The parameters $\kappa_b$ and $f_a$ are related to the measured parameters $B$ and $f$ via $B \approx \sigma \kappa_b/2$ and $f \approx 2F_a/\sigma$. Buckling is induced by steric interaction using a WCA potential (**Abbaspour et al., 2023**) with a V-shaped obstacle. The dynamics of the chain is given by an overdamped Langevin equation and simulated with the molecular dynamics software HOOMD-blue (**Anderson et al., 2020**).

### Critical length

To derive an analytical expression for the critical $\Gamma$ in **Equation 6**, we first solve the homogeneous equation by

$$\kappa_{\omega=0} = c_1 \text{Ai}\left(k^{1/3} r\right) + c_2 \text{Bi}\left(k^{1/3} r\right), \tag{9}$$

with $r = s - 1$, and give a particular solution to the inhomogeneous equation:

$$
\begin{aligned}
\kappa = \quad &\kappa_{\omega=0} + \frac{1}{k} - r^2 \\
&\cdot \Bigg( \; {}_0F_1\left(;\frac{4}{3};\frac{k}{9}r^3\right) \; {}_1F_2\left(\frac{1}{3};\frac{2}{3},\frac{4}{3};\frac{k}{9}r^3\right) \\
&\quad -\frac{1}{2} \; {}_0F_1\left(;\frac{2}{3};\frac{k}{9}r^3\right) \; {}_1F_2\left(\frac{2}{3};\frac{4}{3},\frac{5}{3};\frac{k}{9}r^3\right) \Bigg),
\end{aligned}
\tag{10}
$$

where the ${}_pF_q$ are the generalized hypergeometric functions. The parameters $c_1$ and $c_2$ are determined by the torque boundary conditions. Then, $\Gamma$ is found from the remaining force boundary condition, which boils down to the roots of

$$
\begin{aligned}
&2k \; {}_0F_1\left(;\frac{4}{3};-\frac{k}{9}\right) \; {}_1F_2\left(\frac{1}{3};\frac{2}{3},\frac{4}{3};-\frac{k}{9}\right) \\
&+ {}_0F_1\left(;\frac{2}{3};-\frac{k}{9}\right)\left\{2 - k \; {}_1F_2\left(\frac{2}{3};\frac{4}{3},\frac{5}{3};-\frac{k}{9}\right)\right\} = 2.
\end{aligned}
\tag{11}
$$

The smallest root is $k \approx 30.5722$.

### Acknowledgements

The authors gratefully acknowledge the Algae Culture Collection (SAG) in Göttingen, Germany, for providing the cyanobacteria species *O. lutea* (SAG 1459–3) and *K. animale* (SAG 1459–6), and technical support. We also thank D Strüver, M Benderoth, W Keiderling, and K Hantke for their technical assistance, culture maintenance, and discussions. We further thank D Qian for assistance with the microfluidic devices. The work of L A and S Kl was done within the Max Planck School Matter to Life, supported by the German Federal Ministry of Education and Research (BMBF) in collaboration with the Max Planck Society. We gratefully acknowledge discussions with M Prakash, R Golestanian, A Vilfan, K R Prathyusha, and F Papenfuß.

### Additional information

#### Competing interests
Leila Abbaspour, Stefan Klumpp: The work was done within the Max Planck School Matter to Life, supported by the German Federal Ministry of Education and Research (BMBF) in collaboration with the Max Planck Society. The other authors declare that no competing interests exist.

## Funding

| Funder | Grant reference number | Author |
|---|---|---|
| German Federal Ministry of Education and Research | M526300 | Leila Abbaspour Stefan Klumpp |

The funders had no role in study design, data collection and interpretation, or the decision to submit the work for publication. Open access funding provided by Max Planck Society.

## Author contributions

Maximilian Kurjahn, Conceptualization, Software, Formal analysis, Investigation, Visualization, Methodology, Writing – original draft, Writing – review and editing; Antaran Deka, Software, Formal analysis, Investigation, Methodology, Writing – original draft, Writing – review and editing; Antoine Girot, Resources, Software, Investigation, Methodology, Designed 3-points bending experiments and performed first measurements, Developed corresponding code to analyze data; Leila Abbaspour, Software, Formal analysis, Investigation, Visualization, Methodology, Writing – original draft; Stefan Klumpp, Conceptualization, Supervision, Methodology, Writing – review and editing; Maike Lorenz, Resources, Writing – review and editing; Oliver Bäumchen, Conceptualization, Resources, Supervision, Validation, Investigation, Methodology, Project administration, Writing – review and editing; Stefan Karpitschka, Conceptualization, Software, Formal analysis, Supervision, Validation, Investigation, Methodology, Writing – original draft, Project administration, Writing – review and editing

## Author ORCIDs

Maximilian Kurjahn (ID) http://orcid.org/0000-0002-7395-7327
Leila Abbaspour (ID) https://orcid.org/0000-0003-2546-9748
Stefan Klumpp (ID) https://orcid.org/0000-0003-0584-2146
Stefan Karpitschka (ID) http://orcid.org/0000-0001-9210-2589

Reviewer #1 (Public Review): https://doi.org/10.7554/eLife.87450.3.sa1
Reviewer #2 (Public Review): https://doi.org/10.7554/eLife.87450.3.sa2
Reviewer #3 (Public Review): https://doi.org/10.7554/eLife.87450.3.sa3
Author response https://doi.org/10.7554/eLife.87450.3.sa4

# Additional files

## Supplementary files
• MDAR checklist

## Data availability

All data and relevant source code have been made publicly available here https://doi.org/10.17617/3.QYENUE.

The following dataset was generated:

| Author(s) | Year | Dataset title | Dataset URL | Database and Identifier |
|---|---|---|---|---|
| Kurjahn M | 2023 | Quantifying gliding forces of filamentous cyanobacteria by self-buckling | https://doi.org/10.17617/3.QYENUE | Edmond, 10.17617/3.QYENUE |

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
