## [Editor Report · eLife assessment]

This **valuable** paper describes innovative force measurements of the bending modulus of gliding cyanobacteria, along with measurements of the critical buckling length of the cells, which in combination lead to insight into how these cells produce the force necessary to move. Quantitative analysis **convincingly** shows that the propulsive force and resistive friction coefficient are strongly coupled, which supports propulsion based on adhesion forces rather than slime extrusion.

---

## [Referee Report · Reviewer #1 (Public Review)]

The paper combines experiments on freely gliding cyanobacteria, buckling experiments using two-dimensional V shaped corners, and micropipette force measurements with theoretical models to study gliding forces in these organisms. The aim is to quantify these forces and use the results to perhaps discriminate between competing mechanisms by which these cells move. A large data set of possible collision events are analyzed, bucking events evaluated, and critical buckling lengths estimated. A line elasticity model is used to analyze the onset of buckling and estimate the effective (viscous type) friction/drag that controls the dynamics of the rotation that ensues post-buckling. This value of the friction/drag is compared to a second estimate obtained by consideration of the active forces and speeds in freely gliding filaments. The authors find that these two independent estimates of friction/drag correlate with each other and are comparable in magnitude. The experiments are conducted carefully, the device fabrication is novel, the data set is interesting, and the analysis is solid. The authors conclude that the experiments are consistent with the propulsion being generated by adhesion forces rather than slime extrusion. While consistent with the data, this conclusion is inferred.

Summary:

The paper addresses important questions on the mechanisms driving the gliding motility of filamentous cyanobacteria. The authors aim to understand these by estimating the elastic properties of the filaments, and by comparing the resistance to gliding under (a) freely gliding conditions, and (b) in post-buckled rotational states. Experiments are used to estimate the propulsion force density on freely gliding filaments (assuming over damped conditions). Experiments are combined with a theoretical model based on Euler beam theory to extract friction (viscous) coefficients for filaments that buckle and begin to rotate about the pinned end. The main results are estimates for the bending stiffness of the bacteria, the propulsive tangential force density, the buckling threshold in terms of the length, and estimates of the resistive friction (viscous drag) providing the dissipation in the system and balancing the active force. It is found that experiments on the two bacterial species yield nearly identical value of 𝑓 (albeit with rather large variations). The authors conclude that the experiments are consistent with the propulsion being generated by adhesion forces rather than slime extrusion.

Strengths of the paper:

The strengths of the paper lie in the novel experimental setup and measurements that allow for the estimation of the propulsive force density, critical buckling length, and effective viscous drag forces for movement of the filament along its contour - the axial (parallel) drag coefficient, and the normal (perpendicular) drag coefficient (I assume this is the case, since the post-buckling analysis assumes the bent filament rotates at a constant frequency). These direct measurements are important for serious analysis and discrimination between motility mechanisms.

Weaknesses:

There are aspects of the analysis and discussion that may be improved. I suggest that the authors take the following comments into consideration while revising their manuscript.

The conclusion that adhesion via focal adhesions is the cause for propulsion rather than slime protrusion, is consistent with the experimental results that the frictional drag correlates with propulsion force. At the same time, it is hard to rule out other factors that may result in this (friction) viscous drag - (active) force relationship while still being consistent with slime production. More detailed analysis aiming to discriminate between adhesion vs slime protrusion may be outside the scope of the study, but the authors may still want to elaborate on their inference. It would help if there was a detailed discussion on the differences in terms of the active force term for the focal adhesion-based motility vs the slime motility.

Can the authors comment on possible mechanisms (perhaps from the literature) that indicate how isotropic friction may be generated in settings where focal adhesions drive motility. A key aspect here would probably be estimating the extent of this adhesion patch and comparing it to a characteristic contact area. Can lubrication theory be used to estimate characteristic areas of contact (knowing the radius of the filament, and assuming a height above substrate)? If the focal adhesions typically cover areas smaller than this lubrication area, it may suggest the possibility that bacteria essentially present a flat surface insofar as adhesion is concerned, leading to transversely isotropic response in terms of the drag. Of course, we will still require the effective propulsive force to act along the tangent.

I am not sure why the authors mention that the power of the gliding apparatus is not rate limiting. The only way to verify this would be to put these in highly viscous fluids where the drag of the external fluid comes into the picture as well (if focal adhesions are on the substrate facing side, and the upper side is subject to ambient fluid drag). Also, the friction referred to here has the form of a viscous drag (no memory effect, and thus not viscoelastic or gel-like), and it is not clear if forces generated by adhesion involve other forms of drag such as chemical friction via temporary bonds forming and breaking. In quasi-static settings and under certain conditions such as separation of chemical and elastic time scales, bond friction may yield overall force proportional to local sliding velocities.

For readers from a non-fluids background, some additional discussion of the drag forces, and the forms of friction would help. For a freely gliding filament if 𝑓 is the force density (per unit length), then steady gliding with a viscous frictional drag would suggest (as mentioned in the paper) 𝑓 ∼ 𝑣! 𝐿 𝜂∥. The critical buckling length is then dependent on 𝑓 and on 𝐵 the bending modulus. Here the effective drag is defined per length. I can see from this that if the active force is fixed, and the viscous component resulting from the frictional mechanism is fixed, the critical buckling length will not depend on the velocity (unless I am missing something in their argument), since the velocity is not a primitive variable, and is itself an emergent quantity.

---

## [Referee Report · Reviewer #2 (Public Review)]

In the presented manuscript, the authors first use structured microfluidic devices with gliding filamentous cyanobacteria inside in combination with micropipette force measurements to measure the bending rigidity of the filaments. The distribution of bending rigidities is very broad.

Next, they use triangular structures to trap the bacteria with the front against an obstacle. Depending on the length and rigidity, the filaments buckle under the propulsive force of the cells. The authors use theoretical expressions for the buckling threshold to infer propulsive force, given the measured length and (mean-) stiffnesses. They find nearly identical values for both species, 𝑓 ∼ (1.0 {plus minus} 0.6) nN∕µm, nearly independent of the velocity. These measurements have to be taken with additional care, as then inferred forces depend strongly on the bending rigidity, which already shows a broad distribution.

Finally, they measure the shape of the filament dynamically to infer friction coefficients via Kirchhoff theory. In this section they report a strong correlation with velocity and report propulsive forces that vary over two orders of magnitude.

From a theoretical perspective, not many new results are presented. The authors repeat the the well-known calculation for filaments buckling under propulsive load and arrive at the literature result of buckling when the dimensionless number (f L^3/B) is larger than 30.6 as previously derived by Sekimoto et al in 1995. In my humble opinion, the "buckling theory" section belongs to methods.

Finally, the Authors use molecular dynamics type simulations similar to other models to reproduce the buckling dynamics from the experiments.

Data and source code are available via trusted institutional or third-party repositories that adhere to policies that make data discoverable, accessible and usable.

---

## [Referee Report · Reviewer #3 (Public Review)]

Summary:

This paper presents novel and innovative force measurements of the biophysics of gliding cyanobacteria filaments. These measurements allow for estimates of the resistive force between the cell and substrate and provide potential insight into the motility mechanism of these cells, which remains unknown.

Strengths:

The authors used well-designed microfabricated devices to measure the bending modulus of these cells and to determine the critical length at which the cells buckle. I especially appreciated the way the authors constructed an array of pillars and used it to do 3-point bending measurements and the arrangement the authors used to direct cells into a V-shaped corner in order to examine at what length the cells buckled at. By examining the gliding speed of the cells before buckling events, the authors were able to determine how strongly the buckling length depends on the gliding speed, which could be an indicator of how the force exerted by the cells depends on cell length; however, the authors did not comment on this directly.

Weaknesses:

There are no major weaknesses in the paper.

---

## [Author Response]

**Reviewer 1:**
The paper “Quantifying gliding forces of filamentous cyanobacteria by self-buckling” combines experiments on freely gliding cyanobacteria, buckling experiments using two-dimensional V-shaped corners, and micropipette force measurements with theoretical models to study gliding forces in these organisms. The aim is to quantify these forces and use the results to perhaps discriminate between competing mechanisms by which these cells move. A large data set of possible collision events are analyzed, bucking events evaluated, and critical buckling lengths estimated. A line elasticity model is used to analyze the onset of buckling and estimate the effective (viscous type) friction/drag that controls the dynamics of the rotation that ensues post-buckling. This value of the friction/drag is compared to a second estimate obtained by consideration of the active forces and speeds in freely gliding filaments. The authors find that these two independent estimates of friction/drag correlate with each other and are comparable in magnitude. The experiments are conducted carefully, the device fabrication is novel, the data set is interesting, and the analysis is solid. The authors conclude that the experiments are consistent with the propulsion being generated by adhesion forces rather than slime extrusion. While consistent with the data, this conclusion is inferred.

We thank the reviewer for the positive evaluation of our work.

Summary:The paper addresses important questions on the mechanisms driving the gliding motility of filamentous cyanobacteria. The authors aim to understand these by estimating the elastic properties of the filaments, and by comparing the resistance to gliding under (a) freely gliding conditions, and (b) in post-buckled rotational states. Experiments are used to estimate the propulsion force density on freely gliding filaments (assuming over-damped conditions). Experiments are combined with a theoretical model based on Euler beam theory to extract friction (viscous) coefficients for filaments that buckle and begin to rotate about the pinned end. The main results are estimates for the bending stiffness of the bacteria, the propulsive tangential force density, the buckling threshold in terms of the length, and estimates of the resistive friction (viscous drag) providing the dissipation in the system and balancing the active force. It is found that experiments on the two bacterial species yield nearly identical values of f (albeit with rather large variations). The authors conclude that the experiments are consistent with the propulsion being generated by adhesion forces rather than slime extrusion.

We appreciate this comprehensive summary of our work.

Strengths of the paper:The strengths of the paper lie in the novel experimental setup and measurements that allow for the estimation of the propulsive force density, critical buckling length, and effective viscous drag forces for movement of the filament along its contour – the axial (parallel) drag coefficient, and the normal (perpendicular) drag coefficient (I assume this is the case, since the post-buckling analysis assumes the bent filament rotates at a constant frequency). These direct measurements are important for serious analysis and discrimination between motility mechanisms.

We thank the reviewer for this positive assessment of our work.

Weaknesses:There are aspects of the analysis and discussion that may be improved. I suggest that the authors take the following comments into consideration while revising their manuscript.The conclusion that adhesion via focal adhesions is the cause for propulsion rather than slime protrusion is consistent with the experimental results that the frictional drag correlates with propulsion force. At the same time, it is hard to rule out other factors that may result in this (friction) viscous drag - (active) force relationship while still being consistent with slime production. More detailed analysis aiming to discriminate between adhesion vs slime protrusion may be outside the scope of the study, but the authors may still want to elaborate on their inference. It would help if there was a detailed discussion on the differences in terms of the active force term for the focal adhesion-based motility vs the slime motility.

We appreciate this critical assessment of our conclusions. Of course we are aware that many different mechanisms may lead to similar force/friction characteristics, and that a definitive conclusion on the mechanism would require the combination of various techniques, which is beyond the scope of this work. Therefore, we were very careful in formulating the discussion of our findings, refraining, in particular, from a singular conclusion on the mechanism but instead indicating “support” for one hypothesis over another, and emphasizing “that many other possibilities exist”.

The most common concurrent hypotheses for bacterial gliding suggest that either slime extrusion at the junctional pore complex [A1], rhythmic contraction of fibrillar arrays at the cell wall [A2], focal adhesion sites connected to intracellular motor-microtubule complexes [A3], or modified type-IV pilus apparati [A4] provide the propulsion forces. For the slime extrusion hypothesis, which is still abundant today, one would rather expect an anticorrelation of force and friction: more slime extrusion would generate more force, but also enhance lubrication. The other hypotheses are more conformal to the trend we observed in our experiments, because both pili and focal adhesion require direct contact with a substrate. How contraction of fibrilar arrays would micromechanically couple to the environment is not clear to us, but direct contact might still facilitate force transduction. Please note that these hypotheses were all postulated without any mechanical measurements, solely based on ultra-structural electron microscopy and/or genetic or proteomic experiments. We see our work as complementary to that, providing a mechanical basis for evaluating these hypotheses.

We agree with the referee that narrowing down this discussion to focal adhesion should have been avoided. We rewrote the concluding paragraph (page 8):

“…it indicates that friction and propulsion forces, despite being quite vari able, correlate strongly. Thus, generating more force comes, inevitably, at the expense of added friction. For lubricated contacts, the friction coefficient is proportional to the thickness of the lubricating layer (Snoeijer et al., 2013), and we conjecture active force and drag both increase due to a more intimate contact with the substrate. This supports mechanisms like focal adhesion (Mignot et al., 2007) or a modified type-IV pilus (Khayatan et al., 2015), which generate forces through contact with extracellular surfaces, as the underlying mechanism of the gliding apparatus of filamentous cyanobacteria: more contacts generate more force, but also closer contact with the substrate, thereby increasing friction to the same extent. Force generation by slime extrusion (Hoiczyk and Baumeister, 1998), in contrast, would lead to the opposite behavior: More slime generates more propulsion, but also reduces friction. Besides fundamental fluid-mechanical considerations (Snoeijer et al., 2013), this is rationalized by two experimental observations: i. gliding velocity correlates positively with slime layer thickness (Dhahri et al., 2013) and ii. motility in slime-secretion deficient mutants is restored upon exogenous addition of polysaccharide slime. Still we emphasize that many other possibilities exist. One could, for instance, postulate a regulation of the generated forces to the experienced friction, to maintain some preferred or saturated velocity.”

Can the authors comment on possible mechanisms (perhaps from the literature) that indicate how isotropic friction may be generated in settings where focal adhesions drive motility? A key aspect here would probably be estimating the extent of this adhesion patch and comparing it to a characteristic contact area. Can lubrication theory be used to estimate characteristic areas of contact (knowing the radius of the filament, and assuming a height above the substrate)? If the focal adhesions typically cover areas smaller than this lubrication area, it may suggest the possibility that bacteria essentially present a flat surface insofar as adhesion is concerned, leading to a transversely isotropic response in terms of the drag. Of course, we will still require the effective propulsive force to act along the tangent.

We thank the referee for suggesting to estimate the dimensions of the contact region. Both pili and focal adhesion sites would be of sizes below one micron [A3, A4], much smaller than the typical contact region in the lubricated contact, which is on the order of the filament radius (few microns). So indeed, isotropic friction may be expected in this situation [A5] and is assumed frequently in theoretical work [A6–A8]. Anisotropy may then indeed be induced by active forces [A9], but we are not aware of measurements of the anisotropy of friction in bacterial gliding.

For a more precise estimate using lubrication theory, rheology and extrusion rate of the secreted polysaccharides would have to be known, but we are not aware of detailed experimental characterizations.

We extended the paragraph in the buckling theory on page 5 regarding the assumption of isotropic friction:

“We use classical Kirchhoff theory for a uniform beam of length L and bending modulus B, subject to a force density b⃗ = −f t→− η v→, with an effective active force density f along the tangent t→, and an effective friction proportional to the local velocity v→, analog to existing literature (Fily et al., 2020; Chelakkot et al., 2014; Sekimoto et al., 1995). Presumably, this friction is dominated by the lubrication drag from the contact with the substrate, filled by a thin layer of secreted polysaccharide slime which is much more viscous than the surrounding bulk fluid. Speculatively, the motility mechanism might also comprise adhering elements like pili (Khayatan et al., 2015) or foci (Mignot et al., 2007) that increase the overall friction (Pompe et al., 2015). Thus, the drag due to the surrounding bulk fluid can be neglected (Man and Kanso, 2019), and friction is assumed to be isotropic, a common assumption in motility models (Fei et al., 2020; Tchoufag et al., 2019; Wada et al., 2013). We assume…”

We also extended the discussion regarding the outcome of isotropic friction (page 7):

“…Thus we plot f/v over η in Figure 4 D, finding nearly identical values over about two decades. Since f and η are not correlated with v0, this is due to a correlation between f and η. This relation is remarkable in two aspects: On the one hand, it indicates that friction is mainly isotropic. This suggests that friction is governed by an isotropic process like bond friction or lubrication from the slime layer in the contact with the substrate, the latter being consistent with the observation that mutations deficient of slime secretion do not glide but exogenous addition of slime restores motility (Khayatan et al., 2015). In contrast, hydrodynamic drag from the surrounding bulk fluid (Man and Kanso, 2019), or the internal friction of the gliding apparatus would be expected to generate strongly anisotropic friction. If the latter was dominant, a snapping-like transition into the buckling state would be expected, rather than the continuously growing amplitude that is observed in experiments. On the other hand, it indicates that friction and propulsion forces…”

I am not sure why the authors mention that the power of the gliding apparatus is not rate-limiting. The only way to verify this would be to put these in highly viscous fluids where the drag of the external fluid comes into the picture as well (if focal adhesions are on the substrate-facing side, and the upper side is subject to ambient fluid drag). Also, the friction referred to here has the form of a viscous drag (no memory effect, and thus not viscoelastic or gel-like), and it is not clear if forces generated by adhesion involve other forms of drag such as chemical friction via temporary bonds forming and breaking. In quasi-static settings and under certain conditions such as the separation of chemical and elastic time scales, bond friction may yield overall force proportional to local sliding velocities.

We agree with the referee that the origin of the friction is not easily resolved. Lubrication yields an isotropic force density that is proportional to the velocity, and the same could be generated by bond friction. Importantly, both types of friction would be assumed to be predominantly isotropic. We explicitly referred to lubrication drag because it has been shown that mutations deficient of slime extrusion do not glide [A4].

Assuming, in contrast, that in free gliding, friction with the environment is not rate limiting, but rather the internal friction of the gliding apparatus, i.e., the available power, we would expect a rather different behavior during early-buckling evolution. During early buckling, the tangential motion is stalled, and the dynamics is dominated by the growing buckling amplitude of filament regions near the front end, which move mainly transversely. For geometric reasons, in this stage the (transverse) buckling amplitude grows much faster than the rear part of the filament advances longitudinally. Thus that motion should not be impeded much by the internal friction of the gliding apparatus, but by external friction between the buckling parts of the filament and the ambient. The rate at which the buckling amplitude initially grows should be limited by the accumulated compressive stress in the filament and the transverse friction with the substrate. If the latter were much smaller than the (logitudinal) internal friction of the gliding apparatus, we would expect a snapping-like transition into the buckled state, which we did not observe.

In our paper, we do not intend to evaluate the exact origin of the friction, quantifying the gliding force is the main objective. A linear force-velocity relation agrees with our observations. A detailed analysis of friction in cyanobacterial gliding would be an interesting direction for future work.

To make these considerations more clear, we rephrased the corresponding paragraph on page 7 & 8:

“…Thus we plot f/v over η in Figure 4 D, finding nearly identical values over about two decades. Since f and η are not correlated with v0, this is due to a correlation between f and η. This relation is remarkable in two aspects: On the one hand, it indicates that friction is mainly isotropic. This suggests that friction is governed by an isotropic process like bond friction or lubrication from the slime layer in the contact with the substrate, the latter being consistent with the observation that mutations deficient of slime secretion do not glide but exogenous addition of slime restores motility (Khayatan et al., 2015). In contrast, hydrodynamic drag from the surrounding bulk fluid (Man and Kanso, 2019), or the internal friction of the gliding apparatus would be expected to generate strongly anisotropic friction. If the latter was dominant, a snapping-like transition into the buckling state would be expected, rather than the continuously growing amplitude that is observed in experiments. On the other hand, it indicates that friction and propulsion forces…”

For readers from a non-fluids background, some additional discussion of the drag forces, and the forms of friction would help. For a freely gliding filament if f is the force density (per unit length), then steady gliding with a viscous frictional drag would suggest (as mentioned in the paper) f ∼ v! L η||. The critical buckling length is then dependent on f and on B the bending modulus. Here the effective drag is defined per length. I can see from this that if the active force is fixed, and the viscous component resulting from the frictional mechanism is fixed, the critical buckling length will not depend on the velocity (unless I am missing something in their argument), since the velocity is not a primitive variable, and is itself an emergent quantity.

We are not sure what “f ∼ v! L η||” means, possibly the spelling was corrupted in the forwarding of the comments.

We assumed an overdamped motion in which the friction force density ff (per unit length of the filament) is proportional to the velocity v0, i.e. ff ∼ η v0, with a friction coefficient η. Overdamped means that the friction force density is equal and opposite to the propulsion force density, so the propulsion force density is f ∼ ff ∼ η v0. The total friction and propulsion forces can be obtained by multiplication with the filament length

L, which is not required here. In this picture, v0 is an emergent quantity and f and η are assumed as given and constant. Thus, by observing v0, f can be inferred up to the friction coefficient η. Therefore, by using two descriptive variables, L and v0, with known B, the primitive variable η can be inferred by logistic regression, and f then follows from the overdamped equation of motion.

To clarify this, we revised the corresponding section on page 5 of the paper:

“The substrate contact requires lubrication from polysaccharide slime to enable bacteria to glide (Khayatan et al., 2015). Thus we assume an over- damped motion with co-linear friction, for which the propulsion force f and the free gliding velocity v0 of a filament are related by f = η v0, with a friction coefficient η. In this scenario, f can be inferred both from the observed Lc ∼ (f/B)−1/3 and, up to the proportionality coefficient η, from the observed free gliding velocity. Thus, by combining the two relations, one may expect also a strong correlation between Lc and v0. In order to test this relation for consistency with our data, we include v0 as a second regressor, by setting x = (L−Lc(v0))/∆Lc in Equation 1, with Lc(v0) = (η v0/(30.5722 B))−1/3, to reflect our expectation from theory (see below). Now, η rather than f is the only unknown, and its ensemble distribution will be determined in the regression. Figure 3 E,F show the buckling behavior…”

**Reviewer 2:**
In the presented manuscript, the authors first use structured microfluidic devices with gliding filamentous cyanobacteria inside in combination with micropipette force measurements to measure the bending rigidity of the filaments.Next, they use triangular structures to trap the bacteria with the front against an obstacle. Depending on the length and rigidity, the filaments buckle under the propulsive force of the cells. The authors use theoretical expressions for the buckling threshold to infer propulsive force, given the measured length and stiffnesses. They find nearly identical values for both species, f ∼ (1.0 ± 0.6) nN/µm, nearly independent of the velocity.Finally, they measure the shape of the filament dynamically to infer friction coefficients via Kirchhoff theory. This last part seems a bit inconsistent with the previous inference of propulsive force. Before, they assumed the same propulsive force for all bacteria and showed only a very weak correlation between buckling and propulsive velocity. In this section, they report a strong correlation with velocity, and report propulsive forces that vary over two orders of magnitude. I might be misunderstanding something, but I think this discrepancy should have been discussed or explained.

We regret the misunderstanding of the reviewer regarding the velocity dependence, which indicates that the manuscript should be improved to convey these relations correctly.

First, in the Buckling Measurements section, we did not assume the same propulsion force for all bacteria. The logistic regression yields an ensemble median for Lc (and thus an ensemble median for f), along with the width ∆Lc of the distribution (and thus also the width of the distribution of f). Our result f ∼ (1.0 ± 0.6) nN/µm indicates the median and the width of the distribution of the propulsion force densities across the ensemble of several hundred filaments used in the buckling measurements. The large variability of the forces found in the second part is consistently reflected by this very wide distribution of active forces detected in the logistic regression in the first part.

We did small modifications to the buckling theory paragraph to clarify that in the first part, a distribution of forces rather than a constant value is inferred (page 6)

“Inserting the population median and quartiles of the distributions of bending modulus and critical length, we can now quantify the distribution of the active force density for the filaments in the ensemble from the buckling measurements. We obtain nearly identical values for both species, f ∼ (1.0±0.6) nN/µm, where the uncertainty represents a wide distribution of f across the ensemble rather than a measurement error.”

The same holds, of course, when inferring the distribution of the friction coefficients (page 5):

“The substrate contact requires lubrication from polysaccharide slime to enable bacteria to glide (Khayatan et al., 2015). Thus we assume an over- damped motion with co-linear friction, for which the propulsion force f and the free gliding velocity v0 of a filament are related by f = η v0, with a friction coefficient η. In this scenario, f can be inferred both from the observed Lc ∼ (f/B)−1/3 and, up to the proportionality coefficient η, from the observed free gliding velocity. Thus, by combining the two relations, one may expect also a strong correlation between Lc and v0. In order to test this relation for consistency with our data, we include v0 as a second regressor, by setting x = (L−Lc(v0))/∆Lc in Equation 1, with Lc(v0) = (η v0/(30.5722 B))−1/3, to reflect our expectation from theory (see below). Now, η rather than f is the only unknown, and its ensemble distribution will be determined in the regression. Figure 3 E,F show the buckling behavior…”

The (naturally) wide distribution of force (and friction) leads to a distribution of Lc as well. However, due to the small exponent of 1/3 in the buckling threshold Lc ∼ f 1/3, the distribution of Lc is not as wide as the distributions of the individually inferred f or η. This is visualized in panel G of Figure 3, plotting Lc as a function of v0 (v0 is equivalent to f , up to a proportionality coefficient η). The natural length distribution, in contrast, is very wide. Therefore, the buckling propensity of a filament is most strongly characterized by its length, while force variability, which alters Lc of the individual, plays a secondary role.

In order to clarify this, we edited the last paragraph of the Buckling Measurements section on page 5 of the manuscript:

“…Within the characteristic range of observed velocities (1 − 3 µm/s), the median Lc depends only mildly on v0, as compared to its rather broad distribution, indicated by the bands in Figure 3 G. Thus a possible correlation between f and v0 would only mildly alter Lc. The natural length distribution (Appendix 1—figure 1), however, is very broad, and we conclude that growth rather than velocity or force distributions most strongly impacts the buckling propensity of cyanobacterial colonies. Also, we hardly observed short and fast filaments of K. animale, which might be caused by physiological limitations (Burkholder, 1934).”

Second, in the Profile analysis section, we did not report a correlation between force and velocity. As can be seen in Figure 4—figure Supplement 1, neither the active force nor the friction coefficient, as determined from the analysis of individual filaments, show any significant correlation with the velocity. This is also written in the discussion (page 7):

We see no significant correlation between L or v0 and f or η, but the observed values of f and η cover a wide range (Figure 4 B, C and Figure 4—figure Supplement 1).

Note that this is indeed consistent with the logistic regression: Using v0 as a second regressor did not significantly reduce the width of the distribution of Lc as compared to the simple logistic regression, indicating that force and velocity are not strongly correlated.

In order to clarify this in the manuscript, we modified that part (page 7):

“…We see no significant correlation between L or v0 and f or η, but the observed values of f and η cover a wide range (Figure 4 B,C and Figure 4— figure Supplement 1). This is consistent with the logistic regression, where using v0 as a second regressor did not significantly reduce the width of the distribution of critical lengths or active forces. The two estimates of the friction coefficient, from logistic regression and individual profile fits, are measured in (predominantly) orthogonal directions: tangentially for the logistic regression where the free gliding velocity was used, and transversely for the evolution of the buckling profiles. Thus we plot f/v over η in Figure 4 D, finding nearly identical values over about two decades. Since f and η are not correlated with v0, this is due to a correlation between f and η. This relation is remarkable in two aspects: On the one hand, it indicates that friction is mainly isotropic…”

From a theoretical perspective, not many new results are presented. The authors repeat the well-known calculation for filaments buckling under propulsive load and arrive at the literature result of buckling when the dimensionless number (f L3/B) is larger than 30.6 as previously derived by Sekimoto et al in 1995 [1] (see [2] for a clamped boundary condition and simulations). Other theoretical predictions for pushed semi-flexible filaments [1–4] are not discussed or compared with the experiments. Finally, the Authors use molecular dynamics type simulations similar to [2–4] to reproduce the buckling dynamics from the experiments. Unfortunately, no systematic comparison is performed.[1] Ken Sekimoto, Naoki Mori, Katsuhisa Tawada, and Yoko Y Toyoshima. Symmetry breaking instabilities of an in vitro biological system. Physical review letters, 75(1):172, 1995.[2] Raghunath Chelakkot, Arvind Gopinath, Lakshminarayanan Mahadevan, and Michael F Hagan. Flagellar dynamics of a connected chain of active, polar, brownian particles. Journal of The Royal Society Interface, 11(92):20130884, 2014.[3] Rolf E Isele-Holder, Jens Elgeti, and Gerhard Gompper. Self-propelled worm-like filaments: spontaneous spiral formation, structure, and dynamics. Soft matter, 11(36):7181–7190, 2015.[4] Rolf E Isele-Holder, Julia J¨ager, Guglielmo Saggiorato, Jens Elgeti, and Gerhard Gompper. Dynamics of self-propelled filaments pushing a load. Soft Matter, 12(41):8495–8505, 2016.

We thank the reviewer for pointing us to these publications, in particular the work by Sekimoto we were not aware of. We agree with the referee that the calculation is straight forward (basically known since Euler, up to modified boundary conditions). Our paper focuses on experimental work, the molecular dynamics simulations were included mainly as a consistency check and not intended to generate the beautiful post-buckling patterns observed in references [2-4]. However, such shapes do emerge in filamentous cyanobacteria, and with the data provided in our manuscript, simulations can be quantitatively matched to our experiments, which will be covered by future work.

We included the references in the revision of our manuscript, and a statement that we do not claim priority on these classical theoretical results.

Introduction, page 2:

“…Self-Buckling is an important instability for self-propelling rod-like micro-organisms to change the orientation of their motion, enabling aggregation or the escape from traps (Fily et al., 2020; Man and Kanso, 2019; Isele-Holder et al., 2015; Isele-Holder et al., 2016). The notion of self-buckling goes back to work of Leonhard Euler in 1780, who described elastic columns subject to gravity (Elishakoff, 2000). Here, the principle is adapted to the self-propelling, flexible filaments (Fily et al., 2020; Man and Kanso, 2019; Sekimoto et al., 1995) that glide onto an obstacle. Filaments buckle if they exceed a certain critical length Lc ∼ (B/f)1/3, where B is the bending modulus and f the propulsion force density…”

Buckling theory, page 5:

“…The buckling of gliding filaments differs in two aspects: the propulsion forces are oriented tangentially instead of vertically, and the front end is supported instead of clamped. Therefore, with L < Lc all initial orientations are indifferently stable, while for L > Lc, buckling induces curvature and a resultant torque on the head, leading to rotation (Fily et al., 2020; Chelakkot et al., 2014; Sekimoto et al., 1995). Buckling under concentrated tangential end-loads has also been investigated in literature (de Canio et al., 2017; Wolgemuth et al., 2005), but leads to substantially different shapes of buckled filaments. We use classical Kirchhoff theory for a uniform beam of length L and bending modulus B, subject to a force density b→=−ft→−ηv→, with an effective active force density f along the tangent t→, and an effective friction proportional to the local velocity v→, analog to existing literature (Fily et al., 2020; Chelakkot et al., 2014; Sekimoto et al., 1995)…”

Further on page 6:

“To derive the critical self-buckling length, Equation 5 can be linearized for two scenarios that lead to the same Lc: early-time small amplitude buckling and late-time stationary rotation at small and constant curvature (Fily et al., 2020; Chelakkot et al., 2014 ; Sekimoto et al., 1995). […] Thus, in physical units, the critical length is given by Lc = (30.5722 B/f)1/3, which is reproduced in particle based simulations (Appendix Figure 2) analogous to those in Isele-Holder et al. (2015, 2016).”

Discussion, page 7 & 8:

“…This, in turn, has dramatic consequences on the exploration behavior and the emerging patterns (Isele-Holder et al., 2015, 2016; Abbaspour et al., 2021; Duman et al., 2018; Prathyusha et al., 2018; Jung et al., 2020): (L/Lc)3 is, up to a numerical prefactor, identical to the flexure number (Isele-Holder et al., 2015, 2016; Duman et al., 2018; Winkler et al., 2017), the ratio of the Peclet number and the persistence length of active polymer melts. Thus, the ample variety of non-equilibrium phases in such materials (Isele-Holder et al., 2015, 2016; Prathyusha et al., 2018; Abbaspour et al., 2021) may well have contributed to the evolutionary success of filamentous cyanobacteria.”

**Reviewer 3:**
Summary:This paper presents novel and innovative force measurements of the biophysics of gliding cyanobacteria filaments. These measurements allow for estimates of the resistive force between the cell and substrate and provide potential insight into the motility mechanism of these cells, which remains unknown.

We thank the reviewer for the positive evaluation of our work. We have revised the manuscript according to their comments and detail our replies and modifications next to the individual points below.

Strengths:The authors used well-designed microfabricated devices to measure the bending modulus of these cells and to determine the critical length at which the cells buckle. I especially appreciated the way the authors constructed an array of pillars and used it to do 3-point bending measurements and the arrangement the authors used to direct cells into a V-shaped corner in order to examine at what length the cells buckled at. By examining the gliding speed of the cells before buckling events, the authors were able to determine how strongly the buckling length depends on the gliding speed, which could be an indicator of how the force exerted by the cells depends on cell length; however, the authors did not comment on this directly.

We thank the referee for the positive assessment of our work. Importantly, we do not see a significant correlation between buckling length and gliding speeds, and we also do not see a correlation with filament length, consistent with the assumption of a propulsion force density that is more or less homogeneously distributed along the filament. Note that each filament consists of many metabolically independent cells, which renders cyanobacterial gliding a collective effort of many cells, in contrast to gliding of, e.g., myxobacteria.

In response also to the other referees’ comments, we modified the manuscript to reflect more on the absence of a strong correlation between velocity and force/critical length. We modified the Buckling measurements section on page 5 of the paper:

“The substrate contact requires lubrication from polysaccharide slime to enable bacteria to glide (Khayatan et al., 2015). Thus we assume an over-damped motion with co-linear friction, for which the propulsion force f and the free gliding velocity v0 of a filament are related by f = η v0, with a friction coefficient η. In this scenario, f can be inferred both from the observed Lc ∼ (f/B)−1/3 and, up to the proportionality coefficient η, from the observed free gliding velocity. Thus, by combining the two relations, one may expect also a strong correlation between Lc and v0. In order to test this relation for consistency with our data, we include v0 as a second regressor, by setting x = (L−Lc(v0))/∆Lc in Equation 1, with Lc(v0) = (η v0/(30.5722 B))−1/3, to reflect our expectation from theory (see below). Now, η rather than f is the only unknown, and its ensemble distribution will be determined in the regression. Figure 3 E, F show the buckling behavior…”

Further, we edited the last paragraph of the Buckling measurements section on page 5 of the manuscript:

“Within the characteristic range of observed velocities (1 − 3 µm/s), the median Lc depends only mildly on v0, as compared to its rather broad distribution, indicated by the bands in Figure 3 G. Thus a possible correlation between f and v0 would only mildly alter Lc. The natural length distribution (cf. Appendix 1—figure 1), however, is very broad, and we conclude that growth rather than velocity or force distributions most strongly impacts the buckling propensity of cyanobacterial colonies. Also, we hardly observed short and fast filaments of K. animale, which might be caused by physiological limitations (Burkholder, 1934).”

We also rephrased the corresponding discussion paragraph on page 7:

“…Thus we plot f/v over η in Figure 4 D, finding nearly identical values over about two decades. Since f and η are not correlated with v0, this is due to a correlation between f and η. This relation is remarkable in two aspects: On the one hand, it indicates that friction is mainly isotropic. This suggests that friction is governed by an isotropic process like bond friction or lubrication from the slime layer in the contact with the substrate, the latter being consistent with the observation that mutations deficient of slime secretion do not glide but exogenous addition of slime restores motility (Khayatan et al., 2015). In contrast, hydrodynamic drag from the surrounding bulk fluid (Man and Kanso, 2019), or the internal friction of the gliding apparatus would be expected to generate strongly anisotropic friction. If the latter was dominant, a snapping-like transition into the buckling state would be expected, rather than the continuously growing amplitude that is observed in experiments. On the other hand, it indicates that friction and propulsion forces…”

Weaknesses:There were two minor weaknesses in the paper.First, the authors investigate the buckling of these gliding cells using an Euler beam model. A similar mathematical analysis was used to estimate the bending modulus and gliding force for Myxobacteria (C.W. Wolgemuth, Biophys. J. 89: 945-950 (2005)). A similar mathematical model was also examined in G. De Canio, E. Lauga, and R.E Goldstein, J. Roy. Soc. Interface, 14: 20170491 (2017). The authors should have cited these previous works and pointed out any differences between what they did and what was done before.

We thank the reviewer for pointing us to these references. The paper by Wolgemuth is theoretical work, describing A-motility in myxobacteria by a concentrated propulsion force at the rear end of the bacterium, possibly stemming from slime extrusion. This model was a little later refuted by [A3], who demonstrated that focal adhesion along the bacterial body and thus a distributed force powers A-motility, a mechanism that has by now been investigated in great detail (see [A10]). The paper by Canio et al. contains a thorough theoretical analysis of a filament that is clamped at one end and subject to a concentrated tangential load on the other. Since both models comprise a concentrated end-load rather than a distributed propulsion force density, they describe a substantially different motility mechanism, leading also to substantially different buckling profiles. Consequentially, these models cannot be applied to cyanobacterial gliding.

We included both citations in the revision and pointed out the differences to our work in the introduction (page 2):

“…A few species appear to employ a type-IV-pilus related mechanism (Khayatan et al., 2015; Wilde and Mullineaux, 2015), similar to the better- studied myxobacteria (Godwin et al., 1989; Mignot et al., 2007; Nan et al., 2014; Copenhagen et al., 2021; Godwin et al., 1989), which are short, rod-shaped single cells that exhibit two types of motility: S (social) motility based on pilus extension and retraction, and A (adventurous) motility based on focal adhesion (Chen and Nan, 2022) for which also slime extrusion at the trailing cell pole was earlier postulated as mechanism (Wolgemuth et al., 2005). Yet, most gliding filamentous cyanobacteria do not exhibit pili and their gliding mechanism appears to be distinct from myxobacteria (Khayatan et al., 2015).”

And in Buckling theory, page 5:

“….The buckling of gliding filaments differs in two aspects: the propulsion forces are oriented tangentially instead of vertically, and the front end is supported instead of clamped. Therefore, with L < Lc all initial orientations are indifferently stable, while for L > Lc, buckling induces curvature and a resultant torque on the head, leading to rotation (Fily et al., 2020; Chelakkot et al., 2014; Sekimoto et al., 1995). Buckling under concentrated tangential end-loads has also been investigated in literature (de Canio et al., 2017; Wolgemuth et al., 2005), but leads to substantially different shapes of buckled filaments.”

The second weakness is that the authors claim that their results favor a focal adhesion-based mechanism for cyanobacterial gliding motility. This is based on their result that friction and adhesion forces correlate strongly. They then conjecture that this is due to more intimate contact with the surface, with more contacts producing more force and pulling the filaments closer to the substrate, which produces more friction. They then claim that a slime-extrusion mechanism would necessarily involve more force and lower friction. Is it necessarily true that this latter statement is correct? (I admit that it could be, but is it a requirement?)

We thank the referee for raising this interesting question. Our claim regarding slime extrusion is based on three facts: i. mutations deficient of slime extrusion do not glide, but start gliding as soon as slime is provided externally [A4]. ii. A positive correlation between speed and slime layer thickness was observed in Nostoc [A11]. iii. The fluid mechanics of lubricated sliding contacts is very well understood and predicts a decreasing resistance with increasing layer thickness.

We included these considerations in the revision of our manuscript (page 8):

“…it indicates that friction and propulsion forces, despite being quite variable, correlate strongly. Thus, generating more force comes, inevitably, at the expense of added friction. For lubricated contacts, the friction coefficient is proportional to the thickness of the lubricating layer (Snoeijer et al., 2013), and we conjecture active force and drag both increase due to a more intimate contact with the substrate. This supports mechanisms like focal adhesion (Mignot et al., 2007) or a modified type-IV pilus (Khayatan et al., 2015), which generate forces through contact with extracellular surfaces, as the underlying mechanism of the gliding apparatus of filamentous cyanobacteria: more contacts generate more force, but also closer contact with the substrate, thereby increasing friction to the same extent. Force generation by slime extrusion (Hoiczyk and Baumeister, 1998), in contrast, would lead to the opposite behavior: More slime generates more propulsion, but also reduces friction. Besides fundamental fluid-mechanical considerations (Snoeijer et al., 2013), this is rationalized by two experimental observations: i. gliding velocity correlates positively with slime layer thickness (Dhahri et al., 2013) and ii. motility in slime-secretion deficient mutants is restored upon exogenous addition of polysaccharide slime. Still we emphasize that many other possibilities exist. One could, for instance, postulate a regulation of the generated forces to the experienced friction, to maintain some preferred or saturated velocity.”

Related to this, the authors use a model with isotropic friction. They claim that this is justified because they are able to fit the cell shapes well with this assumption. How would assuming a non-isotropic drag coefficient affect the shapes? It may be that it does equally well, in which case, the quality of the fits would not be informative about whether or not the drag was isotropic or not.

The referee raises another very interesting point. Given the typical variability and uncertainty in experimental measurements (cf. error Figure 4 A), a model with a sightly anisotropic friction could be fitted to the observed buckling profiles as well, without significant increase of the mismatch. Yet, strongly anisotropic friction would not be consistent with our observations.

Importantly, however, we did not conclude on isotropic friction based on the fit quality, but based on a comparison between free gliding and early buckling (Figure 4 D). In early buckling, the dominant motion is in transverse direction, while longitudinal motion is insignificant, due to geometric reasons. Thus, independent of the underlying model, mostly the transverse friction coefficiont is inferred. In contrast, free gliding is a purely longitudinal motion, and thus only the friction coefficient for longitudinal motion can be inferred. These two friction coefficients are compared in Figure 4 D. Still, the scatter of that data would allow to fit a certain anisotropy within the error margins. What we can exclude based on out observation is the case of a strongly anisotropic friction. If there is no ab-initio reason for anisotropy, nor a measurement that indicates it, we prefer to stick with the simplest

assumption. We carefully chose our wording in the Discussion as “mainly isotropic” rather

than “isotropic” or “fully isotropic”.

We added a small statement to the Discussion on page 7 & 8:

“... Thus we plot f/v over η in Figure 4 D, finding nearly identical values over about two decades. Since f and η are not correlated with v0, this is due to a correlation between f and η. This relation is remarkable in two aspects: On the one hand, it indicates that friction is mainly isotropic. This suggests that friction is governed by an isotropic process like bond friction or lubrication from the slime layer in the contact with the substrate, the latter being consistent with the observation that mutations deficient of slime secretion do not glide but exogenous addition of slime restores motility (Khayatan et al., 2015). In contrast, hydrodynamic drag from the surrounding bulk fluid (Man and Kanso, 2019), or the internal friction of the gliding apparatus would be expected to generate strongly anisotropic friction. If the latter was dominant, a snapping-like transition into the buckling state would be expected, rather than the continuously growing amplitude that is observed in experiments. On the other hand, it indicates that friction and propulsion forces ...”

**Recommendations for the authors**

The discussion regarding how the findings of this paper imply that cyanobacteria filaments are propelled by adhesion forces rather than slime extrusion should be improved, as this conclusion seems questionable. There appears to be an inconsistency with a buckling force said to be only weakly dependent on the gliding velocity, while its ratio with the velocity correlates with a friction coefficient. Finally, data and source code should be made publicly available.

In the revised version, we have modified the discussion of the force generating mechanism according to the reviewer suggestions. The perception of inconsistency in the velocity dependence of the buckling force was based on a misunderstanding, as we detailed in our reply to the referee. We revised the corresponding section to make it more clear. Data and source code have been uploaded to a public data repository.

**Reviewer #2 (recommendations for the authors)**

Despite eLife policy, the authors do not provide a Data Availability Statement. For the presented manuscript, data and source code should be provided “via trusted institutional or third-party repositories that adhere to policies that make data discoverable, accessible and usable.” https://elifesciences.org/inside-elife/51839f0a/for-authors-updates- to-elife-s-data-sharing-policiesMost of the issues in this reviewer’s public review should be easy to correct, so I would strongly support the authors to provide an amended manuscript.

We added the Data Availability Statement in the amended manuscript.

References

[A1] E. Hoiczyk and W. Baumeister. “The junctional pore complex, a prokaryotic secretion organelle, is the molecular motor underlying gliding motility in cyanobacteria”. In: Curr. Biol. 8.21 (1998), pp. 1161–1168. doi: 10.1016/s0960-9822(07)00487-3.

[A2] N. Read, S. Connell, and D. G. Adams. “Nanoscale Visualization of a Fibrillar Array in the Cell Wall of Filamentous Cyanobacteria and Its Implications for Gliding Motility”. In: J. Bacteriol. 189.20 (2007), pp. 7361–7366. doi: 10.1128/jb.00706- 07.

[A3] T. Mignot, J. W. Shaevitz, P. L. Hartzell, and D. R. Zusman. “Evidence That Focal Adhesion Complexes Power Bacterial Gliding Motility”. In: Science 315.5813 (2007), pp. 853–856. doi: 10.1126/science.1137223.

[A4] Behzad Khayatan, John C. Meeks, and Douglas D. Risser. “Evidence that a modified type IV pilus-like system powers gliding motility and polysaccharide secretion in filamentous cyanobacteria”. In: Mol. Microbiol. 98.6 (2015), pp. 1021–1036. doi: 10.1111/mmi.13205.

[A5] Tilo Pompe, Martin Kaufmann, Maria Kasimir, Stephanie Johne, Stefan Glorius, Lars Renner, Manfred Bobeth, Wolfgang Pompe, and Carsten Werner. “Friction- controlled traction force in cell adhesion”. In: Biophysical journal 101.8 (2011), pp. 1863–1870.

[A6] Hirofumi Wada, Daisuke Nakane, and Hsuan-Yi Chen. “Bidirectional bacterial gliding motility powered by the collective transport of cell surface proteins”. In: Physical Review Letters 111.24 (2013), p. 248102.

[A7] Jo¨el Tchoufag, Pushpita Ghosh, Connor B Pogue, Beiyan Nan, and Kranthi K Mandadapu. “Mechanisms for bacterial gliding motility on soft substrates”. In: Proceedings of the National Academy of Sciences 116.50 (2019), pp. 25087–25096.

[A8] Chenyi Fei, Sheng Mao, Jing Yan, Ricard Alert, Howard A Stone, Bonnie L Bassler, Ned S Wingreen, and Andrej Kosmrlj. “Nonuniform growth and surface friction determine bacterial biofilm morphology on soft substrates”. In: Proceedings of the National Academy of Sciences 117.14 (2020), pp. 7622–7632.

[A9] Arja Ray, Oscar Lee, Zaw Win, Rachel M Edwards, Patrick W Alford, Deok-Ho Kim, and Paolo P Provenzano. “Anisotropic forces from spatially constrained focal adhesions mediate contact guidance directed cell migration”. In: Nature communications 8.1 (2017), p. 14923.

[A10] Jing Chen and Beiyan Nan. “Flagellar motor transformed: biophysical perspectives of the Myxococcus xanthus gliding mechanism”. In: Frontiers in Microbiology 13 (2022), p. 891694.

[A11] Samia Dhahri, Michel Ramonda, and Christian Marliere. “In-situ determination of the mechanical properties of gliding or non-motile bacteria by atomic force microscopy under physiological conditions without immobilization”. In: PLoS One 8.4 (2013), e61663.